# Non-invasive intradermal imaging of cystine crystals in cystinosis

**Marya Bengali**[1☯], **Spencer Goodman**[1☯], **Xiaoying Sun**[2], **Magdalene A. Dohil**[3], **Ranjan Dohil**[4], **Robert Newbury**[5], **Tatiana Lobry**[1], **Laura Hernandez**[1], **Corinne Antignac**[6,7], **Sonia Jain**[2], **Stephanie Cherqui**[1]*

**1** Division of Genetics, Department of Pediatrics, University of California, San Diego, La Jolla, California, United States of America, **2** Biostatistics Research Center, Herbert Wertheim School of Public Health and Human Longevity Science, University of California, San Diego, La Jolla, California, United States of America, **3** Division of Pediatric Dermatology, Department of Dermatology, Rady Children's Hospital, San Diego, California, United States of America, **4** Division of Pediatric Gastroenterology, Department of Gastroenterology, Rady Children's Hospital, University of California, San Diego, San Diego, California, United States of America, **5** Department of Pathology, Rady Children's Hospital, University of California, San Diego, San Diego, California, United States of America, **6** Laboratory of Hereditary Kidney Diseases, Imagine Institute, Inserm UMR1163, Université de Paris, Paris, France, **7** Department of Molecular Genetics, Necker Hospital, Assistance Publique-Hôpitaux de Paris, Paris, France

☯ These authors contributed equally to this work.

* scherqui@ucsd.edu

## Abstract

### Importance

Development of noninvasive methodology to reproducibly measure tissue cystine crystal load to assess disease status and guide clinical care in cystinosis, an inherited lysosomal storage disorder characterized by widespread cystine crystal accumulation.

### Objective

To develop an unbiased and semi-automated imaging methodology to quantify dermal cystine crystal accumulation in patients to correlate with disease status.

### Design, setting and participants

101 participants, 70 patients and 31 healthy controls, were enrolled at the University of California, San Diego, Cystinosis Clinics, Rady Children's Hospital, San Diego and at the annual Cystinosis Research Foundation family conference for an ongoing prospective longitudinal cohort study of cystinosis patients with potential yearly follow-up.

### Exposures

Intradermal reflectance confocal microscopy (RCM) imaging, blood collection via standard venipuncture, medical record collection, and occasional skin punch biopsies.

**Data Availability Statement:** All data are submitted as supplementary data (Table of patient data).

**Funding:** Stephanie Cherqui received the following funding sources to support this work. 1) National

Institutes of Health (NIH), grant numbers RO1-DK090058 and R01-NS108965. https://urldefense.com/v3/__http://www.nih.gov__;!!LLK065n_VXAQ!0bgSNM3dVDb5-fgGwjiBhx71bGSI2G9GR2oGYLXpURq4hXISFUlHgH5r8-uSxWqDYSo$. 2) California Institute of Regenerative Medicine (CIRM), grant number CLIN-09230. https://urldefense.com/v3/__https://www.cirm.ca.gov/__;!!LLK065n_VXAQ!0bgSNM3dVDb5-fgGwjiBhx71bGSI2G9GR2oGYLXpURq4hXISFUlHgH5r8-uSD78zFm4$. 3) The cystinosis Research Foundation https://www.cystinosisresearch.org/.

**Competing interests:** I have read the journal's policy and the authors of this manuscript have the following competing interests: Stephanie Cherqui is inventor on a patent entitled "Methods of treating mitochondrial disorders" (#20378-201301) and co-inventor on a patent entitled "Methods of treating lysosomal disorders" (#20378-202488). She is a cofounder, shareholder and a member of both the Scientific Board and Board of Directors of Stelios Therapeutics Inc. Stephanie Cherqui also serves as a member of the Scientific Review Board and Board of Trustees of the Cystinosis Research Foundation. The terms of this arrangement have been reviewed and approved by the University of California San Diego in accordance with its conflict of interest policies. This does not alter our adherence to PLOS ONE policies on sharing data and materials. There are no other patents, products in development or marketed products associated with this research to declare.

## Main outcomes and measures

The primary outcome was to establish an automated measure of normalized confocal crystal volume (nCCV) for each subject. Secondary analysis examined the association of nCCV with various clinical indicators to assess nCCV's possible predictive potential.

## Results

Over 2 years, 57 patients diagnosed with cystinosis (median [range] age: 15.1 yrs [0.8, 54]; 41.4% female) were intradermally assessed by RCM to produce 84 image stacks. 27 healthy individuals (38.7 yrs [10, 85]; 53.1% female) were also imaged providing 37 control image stacks. Automated 2D crystal area quantification revealed that patients had significantly elevated crystal accumulation within the superficial dermis. 3D volumetric analysis of this region was significantly higher in patients compared to healthy controls (mean [SD]: 1934.0 $\mu m^3$ [1169.1] for patients vs. 363.1 $\mu m^3$ [194.3] for controls, $P$<0.001). Medical outcome data was collected from 43 patients with infantile cystinosis (media [range] age: 11 yrs [0.8, 54]; 51% female). nCCV was positively associated with hypothyroidism (OR = 19.68, 95% CI: [1.60, 242.46], $P$ = 0.02) and stage of chronic kidney disease (slope estimate = 0.53, 95%CI: [0.05, 1.00], $P$ = 0.03).

## Conclusions and relevance

This study used non-invasive RCM imaging to develop an intradermal cystine crystal quantification method. Results showed that cystinosis patients had increased nCCV compared to healthy controls. Level of patient nCCV correlated with several clinical outcomes suggesting nCCV may be used as a potential new biomarker for cystinosis to monitor long-term disease control and medication compliance.

## Introduction

Cystinosis is an autosomal recessive lysosomal storage disorder (LSD) with a prevalence of 1:100,000 live births [1]. It is characterized by the accumulation of cystine within lysosomes leading to the build-up of crystallized cystine, which is pathognomonic of cystinosis [2, 3]. The disorder is caused by mutations in the *CTNS* gene encoding the lysosomal cystine transporter cystinosin [4, 5].

The most frequent and severe clinical manifestation of cystinosis is the infantile form that initially presents with renal Fanconi syndrome within the first year of life [6, 7]. Patients also present with chronic kidney disease (CKD) leading to end-stage renal failure [8]. Accumulation of soluble cystine and cystine crystals throughout the body progressively causes multi-organ dysfunction such as hypothyroidism, photophobia, neuromuscular disease, and diabetes, ultimately leading to lethality [2, 9]. Cysteamine, an intracellular cystine-depleting agent, is FDA-approved for the treatment of cystinosis, and will reduce the rate of disease progression [9]. Long-term compliance is difficult due to frequent daily dosing, as well as significant side effects including gastrointestinal discomfort, halitosis and body odor [10, 11].

In clinical practice, the efficacy of cysteamine therapy is determined by measuring peripheral white blood cell (WBC) cystine levels [12]. However, this technique is better suited for short-term monitoring and cysteamine dose adjustments but may not reliably reflect tissue

cystine levels. Indeed, not only can a recent dose of cysteamine transiently normalize WBC cystine levels, but also circulating peripheral WBCs may be exposed to higher concentrations of cysteamine [13]. A retrospective study showed that cystine crystal density within intestinal mucosal tissue correlated inversely with duration of cysteamine therapy, estimated glomerular filtration rate (eGFR) and mean WBC cystine levels [14]. However, even though cystine measurement from tissue would be most accurate, it is an invasive approach unsuited to regularly assess disease status and long-term treatment response.

This study develops an alternative method to non-invasively measure long-term cystine crystal accumulation in the skin as a potential novel biomarker for cystinosis. A small study previously showed that cystine crystals could be observed in dermal regions using reflectance confocal microscopy (RCM) [15]. We extended these findings using intradermal RCM to characterize cystine crystal accumulation by developing an automated, unbiased imaging workflow to visualize and quantitate crystallization. Using this methodology, we report that cystinosis patients have increased crystal area and volume compared to healthy controls, with maximal accumulation in the papillary dermis. We compared crystal accumulation to clinical endpoints to find that the normalized confocal crystal volume (nCCV) is significantly associated with CKD stage and hypothyroidism. We have developed a potential new biomarker for cystinosis to non-invasively monitor long-term disease status, which may facilitate routine clinical monitoring.

## Materials and methods

### Study design and collection

Under the UCSD Institutional Review Board (IRB) approved protocol #161168, 30 females and 40 males with cystinosis, with self-reported sex, between the ages of 10 months and 54 years of age, were enrolled with appropriate written informed consent. Parental consent was obtained for patients less than 18 years of age with the appropriate assents for children (ages 7–12 years) and adolescents (ages 13–17 years). Subjects could enroll or leave the study at any time. 17 females and 14 males between the ages of 8 to 85 years of age have similarly been enrolled as healthy control subjects (S1 Table in S1 File). Patients were imaged from 2017 to 2019 at Rady Children's Hospital, the University of California, San Diego, and the Cystinosis Day of Hope conference with optional yearly follow-up. Cystinosis patients and controls who cannot remain still for 5 minutes and/or subjects with highly pigmented skin were excluded from the study, due to the observation of higher backgrounds presenting as false positive crystals [16]. Images acquired from 13 patients and 4 controls in 2017 were used for analytical optimization and excluded from further analysis. A Rady Hospital/UCSD HIPAA authorization form was signed by subjects to allow access to their personal health information (PHI), medical records, for review of diagnosis, medical history and laboratory analysis. Medical outcomes were obtained from records within 6 months before or after confocal imaging.

### Human biopsies for histology and mass spectrometry

To obtain skin tissue, a 4 mm punch biopsy was performed on the area approximately 1–2 cm behind the mastoid region of the right ear. The 4 mm skin punch was cut vertically with half fixed for standard hematoxylin and eosin (H&E) staining and the other half stained for crystal visualization with toluidine blue in glutaraldehyde.

### Intradermal imaging of human skin

We employed the Vivascope 3000 (Caliber I.D.), a handheld *in vivo* RCM device to acquiring 750 μm$^2$ images between the skin surface to superficial collagen. The area ~1–2 cm behind the

earlobe in the mastoid region—selected because of lower sun exposure, minimizing signal background—was disinfected with 70% isopropyl alcohol. Then mineral oil was applied as an immersion fluid. While holding steady, we acquired 78 images with a 2.8 μm step size, roughly 200 μm deep. Several patients were repeatedly imaged at the same location to establish assay reproducibility.

## Image analysis: Automated quantitation of crystal-like structure area and volume

All image processing steps were identically performed automatically on every slice of skin Zstacks using ImagePro Premier 3D (Media Cybernetics) (S1 Fig, Supplementary Methods for full methodology in S1 File). In brief, we first highlighted total crystals by selecting all small bright objects, then separately selected regions of skin structure known as the dermal papillae [17]. Dermal papillae regions were excluded, leaving only isolated crystals. Crystal area was then measured and normalized to the total region of analysis. We generated 3D-reconstructions using these selected regions of interests (ROIs). Total normalized crystal volume (nCCV) was quantified by normalizing the sum volume of crystals against the total region of analysis, yielding a single nCCV value per Zstack. A total of 83 image stacks were acquired from 70 cystinosis patients and 38 from 27 healthy controls.

## ELISAs

The concentration of MCP-1 (Abcam ab179886) and cystatin C (Abcam ab179883) were measured via ELISA using manufacturer protocols from serum collected from cystinosis patients and healthy controls. These assays included 40 samples from cystinosis patients and 24 from healthy controls for Cystatin C, and 41 samples from patients and 25 from controls for MCP1. Samples were run in duplicates.

## *CTNS* mutation genotyping

57 kb deletions were characterized by PCR while exonic Sanger sequencing was employed as previously described to determine smaller *CTNS* indels or substitutions [18] using DNA isolated (Qiagen Cat# 158845) from sterile buccal swabs (Puritan, 25-3406-H).

## Medical records

Estimated Glomerular Filtration Rate (eGFR) was calculated using the bedside Schwartz equation [19]. From patients containing a leukocyte cystine level, granulocyte cystine levels were calculated by multiplying the leukocyte cystine level by 1.95 [20].

## Statistics

2D crystal area was analyzed using two-way ANOVA with Sidak's multiple comparison test to assess the mean differences between patients and controls at every image slice. 3D nCCV and ELISA results were compared between patients and controls using a student's t-test. Receiver Operating Characteristic (ROC) analysis was performed to determine the diagnostic ability of nCCV to discriminate between patients and controls. We employed logistic regression models for binary medical outcomes and linear regression models for continuous outcome comparisons to nCCV. No multiple comparison correction was applied. Statistical significance was evaluated at $P < 0.05$. All analyses were performed using R (https://www.r-project.org) version 3.6.1.

## Results

### Participant characteristics

70 cystinosis patients (median [range] age: 10 yrs [0.8, 54]; 44.6% female) and 31 healthy individuals (38.7 yrs [10, 85]; 53.1% female) were enrolled in the study. Images were acquired from 39 patients and 17 controls in 2018, 44 patients and 21 controls in 2019, and 26 patients and 11 controls in both years (S1 Table in S1 File). Patients typically were diagnosed with cystinosis before the age of 2 (mean [SD] age: 1.82 yrs [2.12]). Except two, all patients are being treated with cysteamine, 76.8% (n = 53) of whom are taking the delayed release twice-daily version (Procysbi$^{TM}$) while 20.3% (n = 14) are taking every 6 hr form (Cystagon$^{TM}$). Of the 70 patients, 21.6% of patients have received at least one kidney transplant. *CTNS* genotypes were obtained from 51 patients, and were in accordance with published rates of causative mutations [21]; 47.1% homozygous for the 57-kb deletion, 39.2% heterozygous, and 13.7% without the 57-kb deletion (S2A Fig in S1 File). Other mutations in *CTNS* gene include indels, frameshifts, substitutions and splice site alterations–all previously reported but two newly identified mutations, c.565C>T -Q189* and c.330-14_331dup. c.565C>T (p.Q189*) and c.330-14_331dup (p. Pro111Glnfs*19). We measured serum MCP-1 and cystatin C levels, and found that cystinosis patients have significantly higher levels of the inflammatory cytokine MCP-1 (mean[SD] patients 283.6 [143.3], controls 206.6 [96.2] pg/ml, $P = 0.021$) as well as renal function marker cystatin C compared to controls (patients 4.39 [1.6], controls 2.29 [0.6], $P < 0.001$) (S2B Fig in S1 File) as previously described [22, 23].

### Observation of tissue cystine crystals by skin biopsy

Skin-punch biopsies were obtained from two cystinosis patients and one healthy control to document dermal abnormalities associated with cystinosis on histopathology using toluidine blue staining. A 20-year-old male diagnosed with infantile cystinosis at 8-months of age exhibited dermal histiocytes containing 2–8 variably shaped intracytoplasmic crystals per cell occupying 25–90% of the cytoplasm (Fig 1A). Similar findings were noted in a 45-year-old male diagnosed with infantile cystinosis at 6-months of age, with periadnexal inflammatory histiocytes containing 4–22 irregularly shaped intracytoplasmic crystals per cell occupying 10–75% of the cytoplasm (Fig 1B). None of these abnormalities were observed in the 29-year-old healthy male control (Fig 1C). H&E staining revealed rare perivascular chronic inflammatory cells in the upper dermis of only the first patient (Fig 1A, right).

### Observation of skin cystine crystals by reflectance confocal microscopy

While cystine crystals were observed in tissue, biopsies are invasive, and thus cannot be performed frequently. We therefore developed a reproducible non-invasive methodology to quantitatively assess cystine crystal accumulation within the skin. We utilized the Vivascope 3000, a RCM device employed to investigate numerous skin pathologies [24, 25], to capture images from the epidermis to the upper dermal region (Fig 1D). This allows for multiple large regions (750 μm$^2$) at various skin layers to be rapidly acquired without the need for fixation or serial sectioning.

In the majority of patients but not controls, we observed multiple small bright irregularly shaped structures (Fig 1D, S1 Video). The region of maximal crystal accumulation was the papillary dermis, the transition zone between the epidermis and dermis [26]. As the papillary dermis is a highly perfused skin layer, comprised of connective tissue containing abundant blood vessels, our findings suggest that cystine crystallization is enhanced by robust vascularization. Unspecific bright structures can occasionally be seen in controls (S2 Video), most

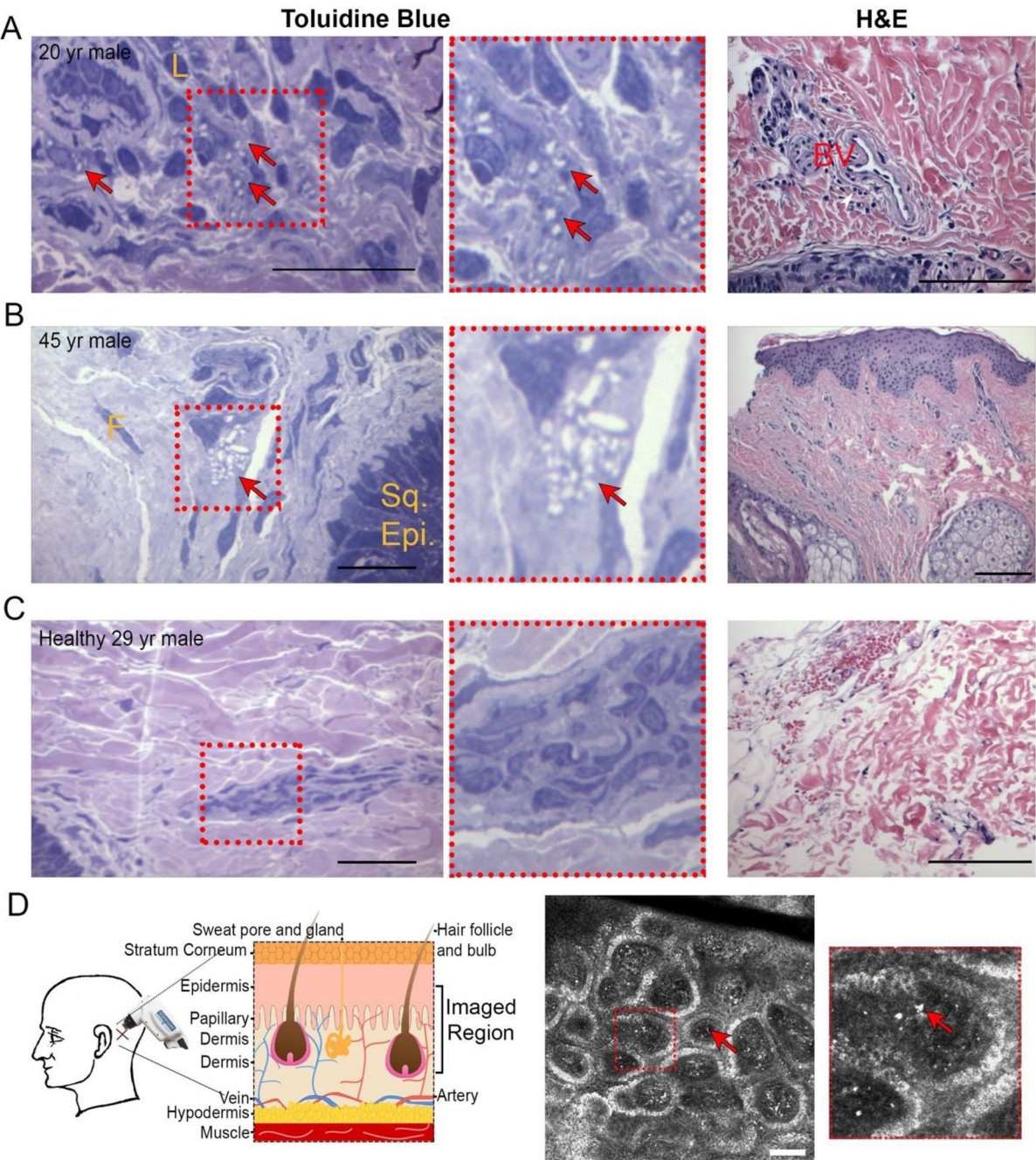

**Fig 1. Skin biopsies and intradermal confocal imaging highlight crystals in cystinosis patients.** A-C, Skin punch biopsies taken from 1–2 cm behind the mastoid region of the right ear of two cystinosis patients (A-B) and a healthy control (C). Skin biopsy samples were sectioned at 1 μm and then stained with Toluidine Blue to detect cystine crystals and H&E for structural evaluation. Arrows indicate crystals. Crystal counting and morphological measurements were conducted by a clinical pathologist. Scale bar toluidine blue = 50 μm. Scale bar H&E = 100 μm. White arrowhead indicate perivascular chronic inflammatory cells in the upper dermis (A). L = lymphocyte, F = fibroblast nucleus, Sq. Epi. = squamous cell epithelium, BV = blood vessel. D, Intradermal imaging methodology and sample patient image. Using a handheld RCM device, a 78 slice Zstack with step size of 2.8 μm was taken behind the left ear beginning within the epidermis. A representative single slice with arrows indicating crystals is shown, while complete patient and control Zstacks are provided in S1 and S2 Videos. Scale bar = 100 μm.

frequently within the superficial layers of the epidermis, especially in darker-skinned subjects, likely due to the presence of melanocytes [16]. This level of background signal remains low throughout all cystinosis patients, and is easily distinguishable from the cystine crystal signal, especially considering that most patients are pale skinned because of defective melanin synthesis [27].

## Crystal quantification by automated image analysis in 2D

In order to rapidly and reproducibly quantitate cystine crystals without bias, we generated a novel image analysis workflow using ImagePro Premier 3D (Media Cybernetics). As described in Methods, for each image slice our macro automatically selects and measures the area of crystal-like structures to generate the sum crystal area (Fig 2A left, S1 Fig in S1 File). Abundant crystal-like structures were detected and quantified in the majority of patients (Fig 2B, S3 Video). Out of the total of 78 slices acquired, significantly higher total crystal area was observed in patients compared to controls within the papillary dermis region, corresponding to slices 15–50 (Fig 2C, S3 Fig in S1 File, S4 Video). The differences were not significant in the epidermis and hypodermis because of high background and low clarity, respectively (S2 Table in S1 File) and were not included in the 3D quantification.

## Crystal quantification by automated image analysis in 3D reconstructions of the papillary dermis

To visualize and quantitate crystal accumulation in each subject as a single value, we generated 3D-reconstructions of the papillary dermis (Fig 2A right). Total normalized confocal crystal volume (nCCV) was assessed by measuring the sum volume of crystals followed by normalization to the total size of the 3D-region excluding skin structure. We report that cystinosis patients have significantly higher nCCV than healthy controls (mean [SD]: 1934.0 $\mu m^3$ [1169.1] for patients vs. 363.1 $\mu m^3$ [194.3] for controls, $P<0.001$) (Fig 2D and 2E, S5 and S6 Videos). ROC analysis showed an area under the curve (AUC) of 0.983 (95%CI: 0.957–1.00). Among cystinosis patients, we also found a significant correlation between age and nCCV (pearson's r = 0.48, p-value$<0.001$), suggesting crystals progressively accumulate over time [9].

To assess the reproducibility of our RCM imaging, the same measured position of the mastoid region in three patients and one control were imaged three times sequentially (S4 Fig in S1 File). We found that all patients and controls had similar measurements—the coefficient of variation (CV) for two patients were below 5% while the third was 18.1%.

## Association between normalized confocal crystal volume and medical outcomes

Medical records from 43 infantile cystinosis patients corresponding to the time of intradermal confocal imaging were collected and blood analysis, urinalysis and disease management data were evaluated in an effort to create a database of cystinosis complications (Table 1, full data from all patients S3 Table in S1 File). As these data were collected from various hospitals, all clinical data were not always available for every individual patient. We compared the current standard measure, WBC cystine level (median [range]: 1.14 [0.53, 6.81] nmol/mg protein ½ cystine), to nCCV and found a moderate correlation (spearman's rho = 0.24, $P = 0.26$). A negative correlation between WBC cystine and cysteamine medication dosage (spearman's rho = -0.44, $P = 0.007$) was observed, but no further significant correlations were detected against clinical outcomes. Granulocyte cystine was only available for a subset of patients (n = 24),

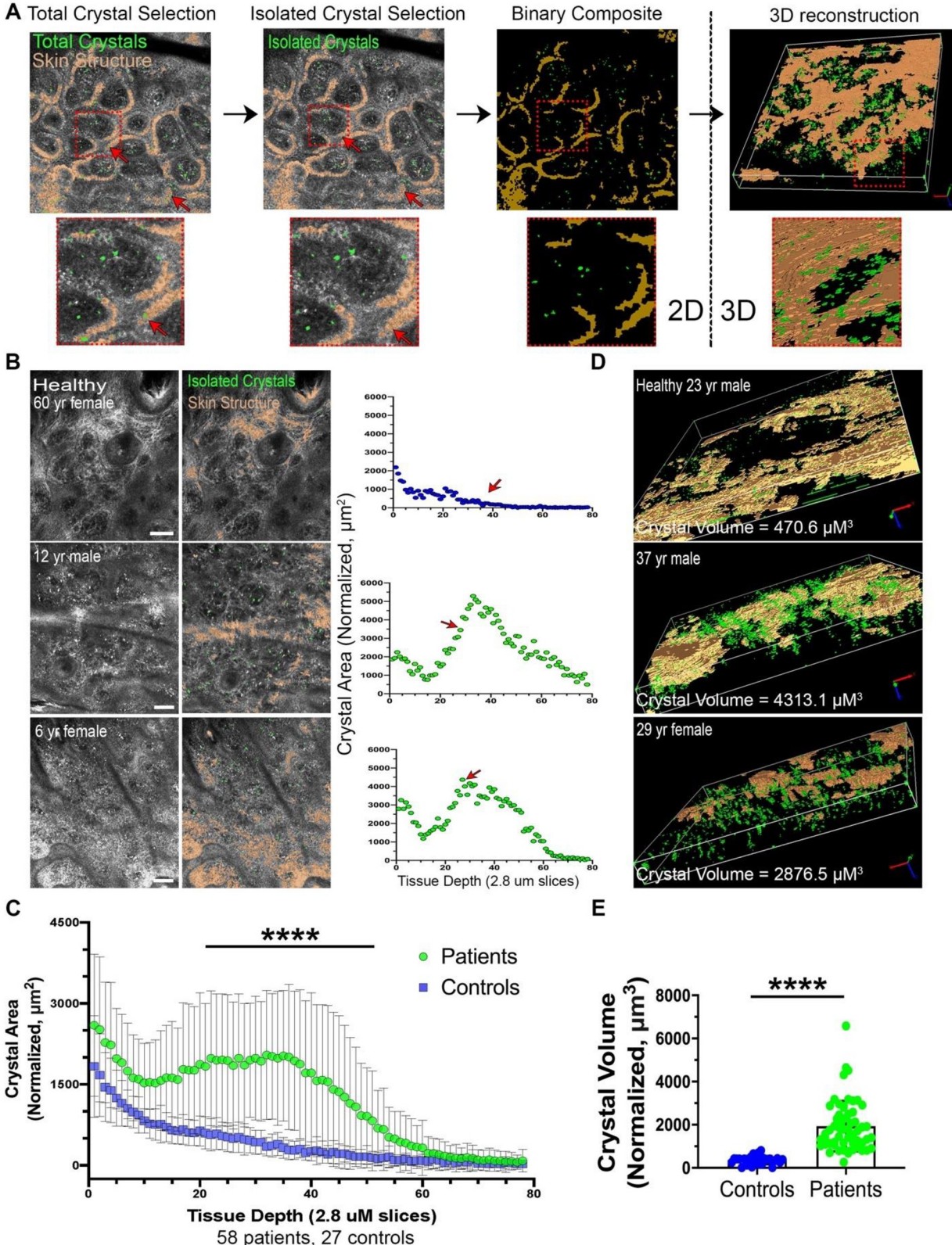

**Fig 2. Automated 2D and 3D image analysis detects an increased crystallization in cystinotic skin.** A total of 83 image stacks were acquired from 70 cystinosis patients and 38 from 27 healthy controls. A, Workflow from initial selection of total crystals and skin structure to final 3D reconstruction of the papillary dermis region. Arrows indicate false-positive crystals which are eliminated due to overlap with skin structure. Full

description of methodology may be found in S1 Fig and Supplementary Methods in (S1 File). B, Representative slices from raw and 2D analyzed intradermal confocal micrographs from a healthy control and patients. XY scatterplot displays the sum of crystal area normalized to total imaged region on the Y-axis vs. tissue depth on X-axis. Arrows indicate which slice is the sample image. C, XY scatterplot depicting mean crystal area +/- SD vs. tissue depth for grouped cystinosis patients vs. healthy controls. For subjects with multiple images, only the most recent encounter was included. Starred region indicates slices where patients have significantly higher crystal area. D, 3D reconstructions of healthy and patient crystal density exclusively in the papillary dermal region due to signal background in epidermis and hypodermis; see Results section "*Crystal quantification by automated image analysis in 2D*" for details. Representative videos are provided in S3 and S4 Videos. E, Boxplot comparing sum crystal volume in papillary dermis (nCCV) +/- SD between patients and controls. **** = $P<0.001$. All scale bars = 100 μm.

suggesting that either the analysis was under-powered or the measure of granulocyte cystine reflected only the short-term impact of cysteamine in the blood as opposed to long-term tissue status. In contrast, we detected significant associations between nCCV and several medical outcomes (Table 2). Higher nCCV was associated with more severe chronic kidney disease stage (slope estimate = 0.53, 95%CI: [0.05, 1.00], $P = 0.03$) after adjusting for age and eGFR. It was also associated with increased risk of hypothyroidism (OR = 19.68, 95%CI: [1.60, 242.46], $P = 0.02$) after adjusting for gender.

## Case studies of cystinosis patients

To take a focused look at the predictive potential of nCCV, this case study examined two patients–Patient 1 who is compliant to cystine-depletion therapy, and heterozygous for the

Table 1. Demographic and clinical characteristics of infantile cystinosis patients.

| Characteristic | N | | Value | Units |
|---|---|---|---|---|
| Age (Median [range]) | 43 | | 11 (0.8–54) | years |
| Gender (%) | | Female | 22 (51.2%) | |
| | | Male | 21 (48.8%) | |
| BMI (Mean [SD]) | 33 | | 17.6 (3.3) | kg/m$^2$ |
| Kidney Transplant (%) | | No | 34 (79.1%) | |
| | | Yes | 9 (20.9%) | |
| Medication (%) | | None | 2 (4.7%) | |
| | | Cystagon | 8 (18.6%) | |
| | | Procysbi | 33 (76.7% | |
| CKD (patients without kidney transplant) (%) | | None | 13 (38.2%) | |
| | | Stage 1 | 4 (11.8%) | |
| | | Stage 2 | 8 (23.5%) | |
| | | Stage 3 | 7 (20.6%) | |
| | | Stage 4 | 2 (5.9%) | |
| Glomerular Filtration Rate (GFR) (Mean [SD]) | 30 | | 66.33 (35.93) | mL/min |
| Hypothyroidism (patients age> = 9) (%) | | No | 12 (60%) | |
| | | Yes | 8 (40%) | |
| Thyroid Stimulating Hormone (TSH) (Mean [SD]) | 33 | | 3.7 (6.11) | IU/L |
| Parathyroid Hormone (PTH) (Mean [SD]) | 28 | | 47.06 (43.56) | pg/mL |
| Cysteamine Dose (Mean [SD]) | 34 | | 1185.32 (694.0) | mg |
| Platelet Count (Mean [SD]) | 38 | | 296.2 (133.34) | 10$^9$/L |
| Serum Sodium (Mean [SD]) | 38 | | 139.21 (2.57) | mmol/L |
| nCCV (Mean [SD]) | 57 | | 1633.5 (806.5) | μm$^3$ |
| Granulocyte cystine level (median [IQR]) | 24 | | 1.07 (0.8–1.5) | nmol/mg |
| Fanconi Syndrome (%) | | No | 10 (23.3%) | |
| | | Yes | 33 (76.7%) | |
| Polydipsia (%) | | No | 27 (64.3%) | |
| | | Yes | 15 (35.7%) | |

**Table 2. Association between nCCV and various medical outcomes.**

| Continuous Medical Outcomes | Coefficient (95% CI) | Wald Test Chi-square | p. value |
|---|---|---|---|
| BMI (n = 33)[a] | 0.45 (-0.66, 1.56) | 0.64 | 0.43 |
| Serum Sodium (n = 38)[a] | -0.72 (-1.55, 0.11) | 2.91 | 0.08 |
| CKD stage (0–4) (n = 22)[c] | 0.53 (0.05, 1.00) | 4.69 | **0.03** |
| Platelet Count (n = 42) | -28.69 (-69.14, 11.76) | 1.93 | 0.16 |
| **Binary Medical Outcomes** | Odds Ratio (95% CI) | Wald Test Chi-square | p. value |
| Hypothyroidism (n = 20)[d] | 19.68 (1.60, 242.46) | 5.41 | **0.02** |
| Fanconi Syndrome (n = 39) | 0.74 (0.36, 1.52) | 0.67 | 0.41 |
| Polydipsia (n = 38) | 0.99 (0.52, 1.88) | 0 | 0.97 |

[a]: Model adjusted for age.

[b]: Model adjusted for age and BMI.

[c]: Model adjusted for age and eGFR and restricted to patients without kidney transplant.

[d]: Model adjusted for gender and restricted to sample with age> = 9.

For each of the continuous medical outcomes, separate linear regression model was fit with standardized nCCV as the main predictor.

For each of the binary medical outcomes, separate logistic regression model was fit with the standardized nCCV as the main predictor.

CKD = Chronic Kidney Disease.

57kb deletion and a one bp duplication (c.696dup- p.Val233Argfs*63) *vs*. Patient 2 who is non-compliant to therapy carrying the 57kb deletion in the homozygous state. Patient 1 is a 20-year-old male presenting with infantile cystinosis diagnosed at 6-month of age, renal Fanconi syndrome, and stage 2 chronic kidney disease (Fig 3A, S7 Video). A medical review revealed the patient was fairly asymptomatic due to cystinosis and had well maintained kidney function. Patient 2 is a 31-year-old male presenting with infantile cystinosis diagnosed at 14-months of age, renal Fanconi syndrome, and chronic kidney failure (Fig 3B, S8 Video). A review of symptoms revealed hypothyroidism, photophobia, diabetes mellitus, hypertension, and dysphagia. The patient has received two kidney transplants at 14- and 26-year-old. Patient 1 had a low and stable nCCV from 2018 (935.24 $\mu m^3$) to 2019 (803.35 $\mu m^3$), while Patient 2 presented with an increase in nCCV between 2019 (3956.3 $\mu m^3$) and 2020 (6586.15 $\mu m^3$) (Fig 3C). Differences in crystallization are further reflected in various clinical indicators between the two patients (Fig 3D).

## Discussion

White blood cell cystine measurements are presently used in cystinosis to monitor disease activity and response to cystine-depletion therapy. However, it remains unclear whether WBC cystine levels accurately reflect tissue cystine levels. The latter is important as it will ultimately dictate the rate of deterioration of organ function and clinical outcome. In our study, a negative correlation between granulocyte cystine and cysteamine medication dosage was shown, but not with clinical outcomes. While sample size limitation may be the cause, peripheral WBC cystine levels are more likely to reflect the most recent dose of cysteamine than the overall tissue cystine storage [13]. In addition, the actual WBC cystine assay is time consuming and expensive with only a few laboratories offering the clinical service. Hence the test, although useful in the day-to-day dosing of cysteamine, is unable to accurately predict long-term response to cystine-depletion therapy.

Measuring cystine in tissues via biopsy would be a more accurate measure of prolonged cystine accretion. However, this is impractical to implement in routine clinical monitoring. However, through histology we did confirm that our new methodology quantitates cystine crystals.

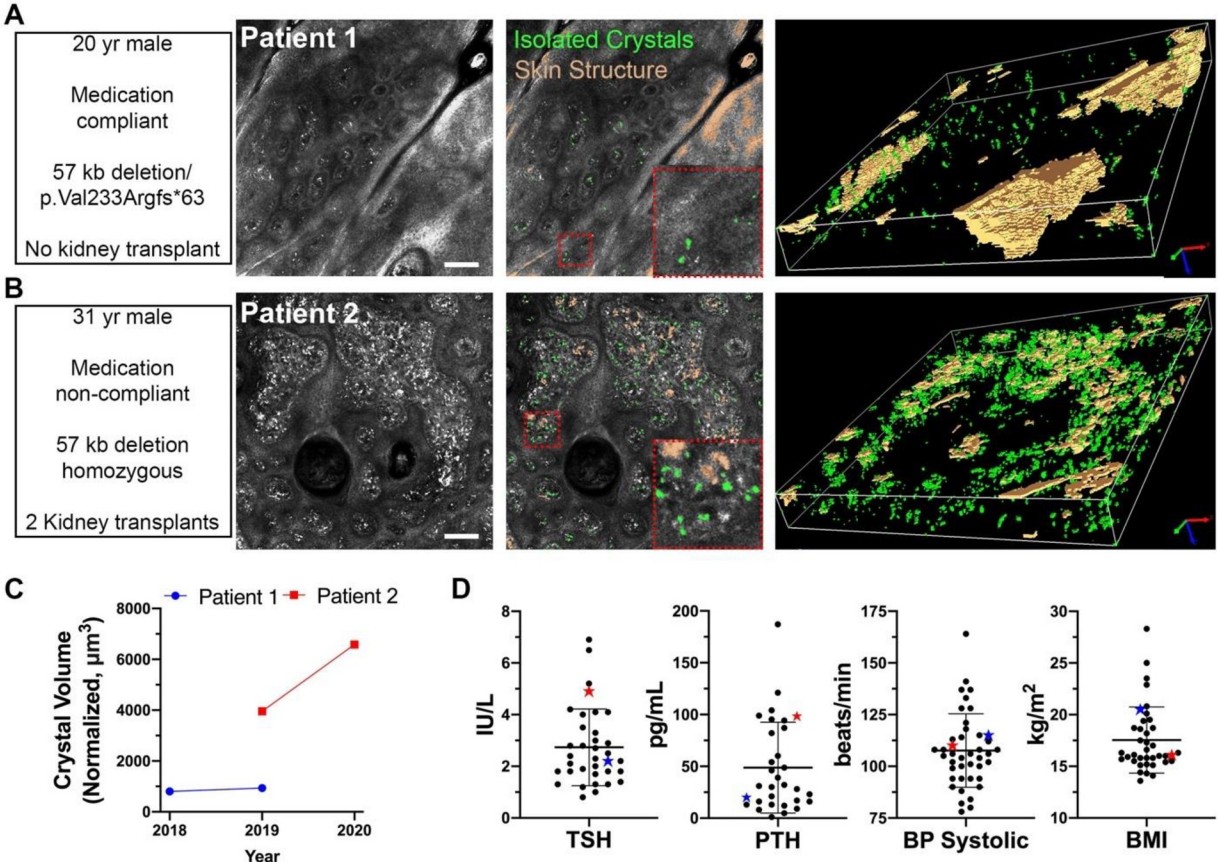

**Fig 3. Focused case study investigating predictive potential of nCCV for two cystinosis patients.** A-B, Intradermal imaging and analysis in 2D and 3D for two cystinosis subjects: Subject 1, a 20 yr-old compliant male without a kidney transplant and compound heterozygous for the 57kb deletion and a frameshift mutation (A), and Subject 2, a 31 yr-old non-compliant patient, bearing the 57kb deletion in the homozygous state, who has received his first kidney transplantation at 14 years and second at 26 years of age (B). Scale bar = 100 μm. C, nCCV quantitation of Patients 1 and 2 for multiple years of acquisition. (D) Selected medical outcomes displaying subject 1 (blue star) and subject 2 (red star) compared to the full set of patients (each dot represents a patient) for various symptoms during the most recent year of acquisition, n = 36 for TSH, n = 31 for PTH, n = 41 for BP Systolic, and n = 37 for BMI.

In contrast, intradermal RCM represents a non-invasive and reliable way to monitor patients regularly to visualize and quantify cystine crystals and determine the impact of oral cysteamine treatment. By using an automated macro for crystal selection that is unbiased and easily reproducible, our results demonstrate standardization, consistency and reproducibility on large cohorts of patients, which can allow future expansion into clinical practice. We thus established an automated measure of normalized confocal crystal volume (nCCV) representing the quantification of crystal volume in 3D reconstituted image stacks. We showed that nCCV was significantly increased in cystinosis patients compared to healthy controls, and that it is significantly and positively correlated with increased age in patients.

We sought to determine if the level of crystal accumulation in the skin would correlate with certain clinical disease outcomes. As expected, due to several factors including medication dosing, compliance to medication and genetic mutation, patients have wide variability in symptom manifestation. The most prevalent long-term complication in cystinosis is CKD [28]. Significant correlation between nCCV and the stage of CKD suggests that level of cystine crystal in the skin may potentially be directly representative of intra-renal cystine accumulation. Thyroid dysfunction, a common endocrine complication in cystinosis, manifests clinically in

patients as hypothyroidism, and biochemically as elevated thyroid stimulating hormone (TSH) but normal T4 levels [29]. We observed a positive association between nCCV and the clinical diagnosis of hypothyroidism. This association strongly suggests that cystine accumulation directly leads to thyroid pathophysiology, a late complication in cystinosis. As expected, due to supportive medications to stabilize TSH, no significant association was observed. In contrast, no correlation was found between nCCV and complications such as renal Fanconi syndrome and polydipsia (Table 2). These symptoms reflect proximal tubulopathy, which has an early onset in cystinosis and is not considered to be caused by cystine accumulation, but rather due to the absence of the cystinosin protein [7]. Similarly, elevated levels of MCP1 and cystatin C are also early markers of the disease so no correlation with nCCV was expected.

Here we show that our nCCV methodology presents the potential to become a biomarker to monitor long-term disease trend, compliance with treatment, and anticipatory guidance for potential complications. Unlike WBC cystine measurement, which is acceptable for evaluating cysteamine dosage adjustments, nCCV directly examines cystine crystal deposition within tissues. The degree of crystallization observed is more likely to be the result of steady prolonged deposition over a longer timeframe, which is more representative of the disease status. nCCV is currently being used as an exploratory endpoint in an ongoing clinical trial to monitor intradermal changes in cystinosis patients after autologous transplantation of gene-modified hematopoietic stem cells (ClinicalTrials.gov Identifier: NCT03897361). Altogether, this study identifies a promising new technology to non-invasively follow disease status and compliance in cystinosis patients.

## Supporting information

**S1 Dataset. 2019 Biostats complete dataset.**
(XLSX)

**S1 File.**
(DOCX)

**S1 Video. Raw intradermal confocal image stacks.** Unprocessed representative raw image stacks acquired from either patient (S1 Video) or healthy control (S2 Video) beginning in the lower epidermis and proceeding across the papillary dermis into the upper dermis region. Images stacks composed of 78 individual slices acquired with a 2.8 μm step size.
(MP4)

**S2 Video. Raw intradermal confocal image stacks.** Unprocessed representative raw image stacks acquired from either patient (S1 Video) or healthy control (S2 Video) beginning in the lower epidermis and proceeding across the papillary dermis into the upper dermis region. Images stacks composed of 78 individual slices acquired with a 2.8 μm step size.
(MP4)

**S3 Video. Selection for isolated crystal and skin structure in 2D.** Sample representative image stacks from patient (S3 Video) and healthy control (S4 Video) processed to select isolated crystals (green) and skin structure (orange) using automated 2D analytical method (see S1 Fig in S1 File).
(MP4)

**S4 Video. Selection for isolated crystal and skin structure in 2D.** Sample representative image stacks from patient (S3 Video) and healthy control (S4 Video) processed to select isolated crystals (green) and skin structure (orange) using automated 2D analytical method (see

S1 Fig in S1 File).
(MP4)

**S5 Video. 3D reconstructions of isolated crystals and skin structure.** Reconstruction of isolated crystals and skin structure in the papillary dermis skin region. Sample representative 3D reconstructions of region of maximal crystal accumulation in the patient (S5 Video) and healthy control (S6 Video). Isolated crystals and skin structure are displayed in green and orange, respectively. Reconstruction built using binary composite of ROI detected via 2D analysis from slices 15–50–42–140 μm from start of image acquisition. See S1 Fig in (S1 File), steps 6–7 for details.
(MP4)

**S6 Video. 3D reconstructions of isolated crystals and skin structure.** Reconstruction of isolated crystals and skin structure in the papillary dermis skin region. Sample representative 3D reconstructions of region of maximal crystal accumulation in the patient (S5 Video) and healthy control (S6 Video). Isolated crystals and skin structure are displayed in green and orange, respectively. Reconstruction built using binary composite of ROI detected via 2D analysis from slices 15–50–42–140 μm from start of image acquisition. See S1 Fig in (S1 File), steps 6–7 for details.
(MP4)

**S7 Video. 3D reconstructions of case study patient images.** Reconstructions as previously described of isolated crystals and skin structure from Patient 1 (S7 Video) and Patient 2 (S8 Video).
(MP4)

**S8 Video. 3D reconstructions of case study patient images.** Reconstructions as previously described of isolated crystals and skin structure from Patient 1 (S7 Video) and Patient 2 (S8 Video).
(MP4)

## Acknowledgments

We thank Dr. Antoanella Calame for her input on dermal layer pathology. We gratefully acknowledge the contributions of the laboratory of Dr. Bruce Barshop and Jon Gangoiti for their invaluable aid in cystine quantification by mass spectrometry. We would like to thank all the cystinosis patients and healthy controls that volunteered to participate to this study. This work was supported and funded by the Cystinosis Research Foundation as well as the National Institute of Health (NIH) RO1-DK090058 and R01-NS108965, the California Institute of Regenerative Medicine (CIRM, CLIN-09230).

## Author Contributions

**Conceptualization:** Marya Bengali, Spencer Goodman, Ranjan Dohil, Stephanie Cherqui.

**Data curation:** Marya Bengali, Spencer Goodman.

**Formal analysis:** Marya Bengali, Spencer Goodman, Xiaoying Sun, Robert Newbury, Corinne Antignac, Sonia Jain, Stephanie Cherqui.

**Funding acquisition:** Stephanie Cherqui.

**Methodology:** Magdalene A. Dohil, Sonia Jain.

**Project administration:** Stephanie Cherqui.

**Resources:** Stephanie Cherqui.

**Supervision:** Stephanie Cherqui.

**Validation:** Laura Hernandez, Sonia Jain.

**Visualization:** Magdalene A. Dohil, Tatiana Lobry.

**Writing – original draft:** Marya Bengali, Spencer Goodman.

**Writing – review & editing:** Magdalene A. Dohil, Ranjan Dohil, Robert Newbury, Corinne Antignac, Sonia Jain.

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
