## [Decision Letter · Decision Letter 0]

14 Jan 2021

PONE-D-20-31230

Non-invasive intradermal imaging of cystine crystals in cystinosis

PLOS ONE

Dear Dr. Cherqui,

Thank you for submitting your manuscript to PLOS ONE. After careful consideration, we feel that it has merit but does not fully meet PLOS ONE’s publication criteria as it currently stands. Therefore, we invite you to submit a revised version of the manuscript that addresses the points raised during the review process.

We look forward to receiving your revised manuscript.

Kind regards,

Thomas Abraham, PhD

Academic Editor

PLOS ONE

Journal Requirements:

3.We note that you have a patent relating to material pertinent to this article. Please provide an amended statement of Competing Interests to declare this patent (with details including name and number), along with any other relevant declarations relating to employment, consultancy, patents, products in development or modified products etc. Please confirm that this does not alter your adherence to all PLOS ONE policies on sharing data and materials, as detailed online in our guide for authors http://journals.plos.org/plosone/s/competing-interests by including the following statement: "This does not alter our adherence to  PLOS ONE policies on sharing data and materials.” If there are restrictions on sharing of data and/or materials, please state these. Please note that we cannot proceed with consideration of your article until this information has been declared.

5. Please upload a copy of Supporting Information eVideos 1-2., eVideos 3-4., eVideos 5-6. and eVideos 7-8.which you refer to in your text on page 26.

6. Please upload a copy of eFigure 1, eFigure 2A, eFigure 3 and eFigure 4  to which you refer in your text on page 7, 9, 11 and 12. If the figure is no longer to be included as part of the submission please remove all reference to it within the text.

Reviewers' comments:

Reviewer's Responses to Questions

**Comments to the Author**

1. Is the manuscript technically sound, and do the data support the conclusions?

Reviewer #1: Yes

2. Has the statistical analysis been performed appropriately and rigorously? 

Reviewer #1: Yes

3. Have the authors made all data underlying the findings in their manuscript fully available?

Reviewer #1: Yes

4. Is the manuscript presented in an intelligible fashion and written in standard English?

Reviewer #1: Yes

5. Review Comments to the Author

Reviewer #1: This study highlights promising new technology to non-invasively follow disease status in cystinosis.

There are no previous studies done with this technology for cystinosis patient. Manuscript is technically sound and data support the conclusion but below are some comments:

1. Include final sample sizes used in statistics section within methods. For example it was not clear what sample size was used for ELISA? Mention if replicates were performed?

2. Point out in H&E staining image perivascular chronic inflammatory cells in figure 1A as mentioned in page 10.

3. Figure 2 C states that “Preliminary slices were excluded due to signal background these

skin layers in healthy controls.” This statement is not clear, clarify this statement further, What is the tissue depth of these excluded slices? Were they present in the starred region? What is justification to remove them? Will this effect results if we include them, if yes is there background in patients’ images as well at this location? By excluding this are we still comparing pateints vs controls in similar experimental setup/ conditions?

4. Indicate sample size used for figure 3D

5. Typographical error in page 14 second paragraph line 3. Correct hrough to “Through”

6. Conclusion is based on results observed but can this methodology be applied to people with with dark skin people? Are there any other challenges with this methodology?

6. PLOS authors have the option to publish the peer review history of their article (what does this mean?). If published, this will include your full peer review and any attached files.

Reviewer #1: No

---

## [Author Response · Author response to Decision Letter 0]

25 Jan 2021

Response to Academic Editor:

 Response: we formatted the revised manuscript to meet PLOS ONE's style requirements.

Response: This information has been added in the Methods section page 6.

3.We note that you have a patent relating to material pertinent to this article. Please provide an amended statement of Competing Interests to declare this patent (with details including name and number), along with any other relevant declarations relating to employment, consultancy, patents, products in development or modified products etc. Please confirm that this does not alter your adherence to all PLOS ONE policies on sharing data and materials, as detailed online in our guide for authors http://journals.plos.org/plosone/s/competing-interests by including the following statement: "This does not alter our adherence to PLOS ONE policies on sharing data and materials.” If there are restrictions on sharing of data and/or materials, please state these. Please note that we cannot proceed with consideration of your article until this information has been declared.

Response: I updated my disclosure with all my patent information. However, these patents are not related to this article. The patent entitled “Methods of treating mitochondrial disorders” and “Methods of treating lysosomal disorders” are related to treating mitochondrial and lysosomal diseases, respectively, with gene-corrected hematopoietic stem cells. The present study reports a new technology to non-invasively follow-up the disease status of patients affected with cystinosis. There is no patent for this present study so no restriction for sharing the data or materials.

Response: The ethics statement only appears in the Methods section.

5. Please upload a copy of Supporting Information eVideos 1-2., eVideos 3-4., eVideos 5-6. and eVideos 7-8.which you refer to in your text on page 26.

Response: While we thought these files were uploaded, we apologize if they were not. The Supporting Information eVideos, now renamed as S# Videos, have been uploaded. 

6. Please upload a copy of eFigure 1, eFigure 2A, eFigure 3 and eFigure 4 to which you refer in your text on page 7, 9, 11 and 12. If the figure is no longer to be included as part of the submission please remove all reference to it within the text.

Response: While we thought these files were uploaded, we apologize if they were not. The Supporting Information file containing the eFigures, now renamed as S# Figures, has been uploaded. 

Response to Reviewers

This study highlights promising new technology to non-invasively follow disease status in cystinosis. 

There are no previous studies done with this technology for cystinosis patient. Manuscript is technically sound and data support the conclusion but below are some comments: 

Response: We acknowledge the reviewer for his comments.

1. Include final sample sizes used in statistics section within methods. For example it was not clear what sample size was used for ELISA? Mention if replicates were performed? 

Response: We added this information in the ELISA section in the Methods page 7. Also, this information also appears in the S2 Figure legend. Finally, in the Supporting Information, which seems to be missing in the final submission, there is the S1 Table that details the numbers of patients and controls enrolled and analyzed by confocal microscope and the number of medical records obtained.

2. Point out in H&E staining image perivascular chronic inflammatory cells in figure 1A as mentioned in page 10. 

Response: Arrowhead has been added in Figure 1A to point out the perivascular chronic inflammatory cells.

3. Figure 2 C states that “Preliminary slices were excluded due to signal background these skin layers in healthy controls.” _This statement is not clear, clarify this statement further, What is the tissue depth of these excluded slices? Were they present in the starred region? What is justification to remove them? Will this effect results if we include them, if yes is there background in patients_’ _images as well at this location? By excluding this are we still comparing pateints vs controls in similar experimental setup/ conditions? 

Response: We thank the reviewer to have pointed out this sentence, which was actually a mistake as we did not exclude any slice for the 2D quantification. Indeed, as stipulated in the Result section “Crystal quantification by automated image analysis in 2D”, out of the 78 slices that were captured per patient, significantly higher total crystal area was observed in patients compared to controls within the papillary dermis region, corresponding to slices 15-50 (Fig 2C, S3 Fig., S4 Video). The differences were not significant in the epidermis and hypodermis because of high background and low clarity, respectively (S2 Table). Therefore, for the 3D quantification we used only the papillary dermis region which present the least artifacts and background. We corrected the Figure 2 legend accordingly.

4. Indicate sample size used for figure 3D 

Response: Each patient is represented by a dot; we added this information in the legend of figure 3D as well as sample size for each medical outcome. This information is also available in the complete dataset supplementary Table “2019 Biostasts Complete Dataset”.

5. Typographical error in page 14 second paragraph line 3. Correct hrough to “Through” _

Response: This typo has been corrected.

6. Conclusion is based on results observed but can this methodology be applied to people with with dark skin people? Are there any other challenges with this methodology? 

Response: As stipulated in the manuscript, people with dark skin will present with higher background level, especially in the epidermis and hypodermis area. This is indeed a limitation of this technology. However, for the 3D quantification of the cystine crystal, we use only the papillary dermis region, which mitigate this issue. In addition, this technology is specific for imaging and quantifying cystine crystal in the skin, which can be found only in cystinosis patients who have very pale skin due to deficiency in melanin production [1]. Therefore, this issue is very limited for this particular population. We made this point clearer in the results section page 11. 

1. Chiaverini C, Sillard L, Flori E, Ito S, Briganti S, Wakamatsu K, et al. Cystinosin is a melanosomal protein that regulates melanin synthesis. FASEB journal : official publication of the Federation of American Societies for Experimental Biology. 2012;26(9):3779-89. doi: 10.1096/fj.11-201376. PubMed PMID: 22649030.

---

## [Editor Report · Decision Letter 1]

15 Feb 2021

Non-invasive intradermal imaging of cystine crystals in cystinosis

PONE-D-20-31230R1

Dear Dr. Cherqui,

We’re pleased to inform you that your manuscript has been judged scientifically suitable for publication and will be formally accepted for publication once it meets all outstanding technical requirements.

Kind regards,

Thomas Abraham, PhD

Academic Editor

PLOS ONE
---

## [Editor Report · Acceptance letter]

23 Feb 2021

PONE-D-20-31230R1 

Non-invasive intradermal imaging of cystine crystals in cystinosis 

Dear Dr. Cherqui:

I'm pleased to inform you that your manuscript has been deemed suitable for publication in PLOS ONE. Congratulations! Your manuscript is now with our production department. 

Kind regards, 

on behalf of

Dr. Thomas Abraham 

Academic Editor

PLOS ONE